# Modification of Cotton Fabric with Molecularly Imprinted Polymer-Coated Carbon Dots as a Sensor for 17 α-methyltestosterone

**DOI:** 10.3390/molecules27217257

**Published:** 2022-10-26

**Authors:** Monyratanak Lim, Sudtida Pliankarom Thanasupsin, Nisakorn Thongkon

**Affiliations:** 1Department of Chemistry, Faculty of Science, King Mongkut’s University of Technology Thonburi, Bangkok 10140, Thailand; 2Chemistry for Green Society and Healthy Living Research Unit, Faculty of Science, King Mongkut’s University of Technology Thonburi, Bangkok 10140, Thailand

**Keywords:** molecularly imprinted polymers, cotton fabric, carbon dots, 17 α-methyltestosterone, fluorescence image, smartphone detection

## Abstract

Molecularly imprinted polymers@ethylenediamine-modified carbon dots grafted on cotton fabrics (MIPs@EDA-CDs/CF) and smartphone-based fluorescence image analysis were proposed and used for the first time for the detection of 17 α-methyltestosterone (MT). The EDA-CDs were synthesized and grafted on cotton fabric before coating with the MIPs. The MIPs were synthesized using the MT as a template molecule, methacrylic acid (MAA) as a functional monomer, ethylene glycol dimethacrylate (EGDMA) as a cross-linker, and azobisisobutyronitrile (AIBN) as an initiator. The MIPs@EDA-CDs/CF were characterized using FTIR, SEM-EDS, and RGB fluorescence imaging. The fluorescence images were also taken using a smartphone and the ImageJ program was used for RGB measurement. The Δ red intensity was linearly proportional to MT concentration in the range of 100 to 1000 μg/L (R^2^ = 0.999) with a detection limit of 44.4 μg/L and quantification limit of 134 μg/L. The MIPs@EDA-CDs/CF could be stored at 4 °C for a few weeks and could be reused twice. The proposed method could apply for the specific determination of MT in water and sediment samples along with satisfactory recoveries of 96–104% and an acceptable relative standard deviation of 1–6% at the ppb level.

## 1. Introduction

Endocrine-disrupting chemicals (EDCs) have been released into the environment in large numbers and quantities since World War II. EDCs are substances or chemical mixtures that interfere with the function of the human body’s hormones [1]. EDCs are compounds that can be found in the environment, food, pesticides, skin care, and manufactured products (some plastic bottles and containers, toys, food cans, etc.). Since EDCs come from many different sources, people are exposed to EDCs through the air, food, and water. EDCs can also enter the skin and body through personal care products. Even a modest amount of EDC consumption can result in hormonal imbalance, especially in children. Sometimes they are stored in body fats and transferred to the developing fetus via the placenta [2]. In general, well-known active EDCs found in the environment are members of the chemical group of steroid hormones, which are formed naturally or synthesized by humans. MT, Bisphenol A (BPA), Perchlorate, Dioxins, Triclosan, Polychlorinated biphenyls (PCBs), Perfluorochemicals (PFCs), Nonylphenol (NP), and Dichlorodiphenyldichloroethane (DDT) and its metabolite Dichlorodiphenyl-dichloroethylene (DDE) are classified as EDCs [2,3].

MT, a synthetic hormone that binds to the androgen receptor, is known for interfering effects on the endocrine systems of invertebrate and vertebrate organisms [4]. It is a synthetic steroid that can induce significant biological adverse effects such as liver damage, embryotoxicity, and cancer because of its slow metabolism and high bioaccumulation in the body [5]. Nevertheless, MT is still widely used to cultivate all-male fish production in Thailand, resulting in high reproductive efficiency, quicker male development, and more standard size than females without concern about contamination of the residue MT in ponds [6,7]. The residue MT from uneaten and leftover MT-impregnated fish is discharged into the aquatic environmental system during the process of cleaning the ponds [8]. Ponds for raising tilapia in Thailand are mostly made of clay or cement. Sediments from the ponds are often dumped on surrounding areas [9] which could pose risks to downstream aquatic ecosystems or soil around the pond [10]. In Thailand, the production of tilapia, about 200,000 tons per year, is very important in the agricultural industry. Therefore, MT has been widely used in commercial farms for all-male production [11]. In consequence, water and sediment contamination by MT has increased in recent years as fish farming production has increased. However, people are unaware that the aquatic environment has undergone hormone-induced sex changes. Moreover, the long-term effects on health are unknown [12] ranging from acute to chronic diseases [13], such as breast enlargement, headache, anxiety, depressed mood, decreasing sperm count, causing reproductive abnormalities, and increasing or decreasing interest in sex [3,14].

Several methods combined with analytical instruments including gas chromatography–mass spectrometry (GC–MS) [15], high-performance liquid chromatography (HPLC) [16], UV–visible spectrophotometry [17], high-performance liquid chromatography–mass spectrometry (HPLC–MS) [18], liquid chromatography–mass spectrometry (LC–MS) [19], chemiluminescence [20], and capillary electrophoresis [21] have been performed for quantitative analysis of MT in various samples. However, these methods require complicated, expensive, time-consuming, and large instruments in the laboratory with skillful operators. Since smartphones have the potential to be more accessible and less expensive than analytical laboratory instruments, they are now equivalent to miniature computers completed with operating systems, internal memory, and a high-resolution camera. The smartphone camera has been used in various colorimetric sensor detection, including enzyme-linked immunosorbent assays, lateral flow immunoassays, and nanoparticle-based homogeneous assays [22,23]. In addition to these analytical techniques, combining an optical cotton sensor based on molecularly imprinted polymers (MIPs)-coated carbon dots with a smartphone-based fluorescence image is a promising approach since it is a feasible alternative method for onsite analysis with high sensitivity and selectivity.

Cotton fabrics, a renewable bioresource, are primarily made up of the most abundant biopolymer cellulose with some noncellulosic components surrounding the cellulose core, such as hydrophobic impurities (waxes, pectin, and proteins) [24,25]. They are inexpensive, simple to use, portable, and easily modified with chemical reagents to change their functionalities. Therefore, they have been investigated for their applications in biomedicine, magnetics, sensors, and other devices [26]. Functional cotton fabric modification is one way to add value and satisfy rising research needs. The abundant hydroxyl groups on the cotton’s surface provide active sites for functionalization [27].

CDs have received much attention in recent decades. These particles are synthesized from carbon materials and are one of the newest members of the quantum nanomaterials family. Since the discovery of CDs, these nanomaterials have received a lot of attention because of their superior properties, including easy functionalization, good photostability, excellent water solubility, bright fluorescence emission, tunable surface functionalities, great and green biocompatibility, low toxicity, and easy synthetic routes [28,29] with various synthesized techniques [30,31]. In this work, CDs were synthesized using a microwave-assisted method, which was a simple and uncomplicated process with numerous advantages such as energy savings, environmental friendliness, and high quantum yield [32]. In the past few years, MIPs have been used along with carbon dots to improve the sensitivity, selectivity, and anti-interference ability of CDs-based sensors [33]. Because of the importance of sensing applications, the MIPs@EDA-CDs/CF-based fluorescent sensing method was prepared as described in this work.

Molecular imprinting technology is based on the principle of enzymes and substrates, hormones and receptors, and antibodies and antigens, which are specific to the recognition of the target analytes [34]. MIPs can be synthesized using different types and combinations of monomers, cross-linkers, solvents, and initiators for numerous fabrications, depending on analytes as template molecules and their applications [35,36]. The functional groups in the monomer can be formed around the template molecules in a pre-polymerization which is locked in its conformation during cross-linking polymerization through non-covalent or covalent interactions. After that, the template molecules are removed from the binding sites, leaving the specific cavities and vacant recognition sites which are structurally similar to the original template [37,38,39,40,41,42,43]. However, the complete removal of the templates from the MIPs can be incomplete or difficult to achieve despite extensive washing steps and prolonged processes with large volumes of desorption solvent. In the majority of the works, non-imprinted polymers (NIPs) were synthesized under identical conditions to the MIPs, except that the templates were absent along with polymerization [44].

Several techniques of synthesizing MIPs for target analytes have been reported [38,45,46,47]. Nevertheless, very few reports were available on MT templating using the molecular imprinting technique. Most research works in the literature were related to the preparation of the MIPs as an adsorbent for batch adsorption studies [5], solid phase extraction [48], and the stationary phase of the HPLC column [18]. The prepared MIPs were synthesized using precipitation polymerization, which requires many variables to control the size, porosity, and surface area of these particles for their adsorption ability. Compared with the solid MIPs, the modified cotton fabrics were more portable and simpler to use for the first application of MT detection. In this work, the cotton fabric is modified with MIPs coated on CDs as a sensor (MIPs@EDA-CDs/CF) for MT detection. EDA-CDs, synthesized by citric acid (CA) and EDA as sources of carbon and amino functional groups, respectively, were grafted on the cotton fabric, and MIPs were synthesized using MT as a template molecule, MAA as a functional monomer, EGDMA as a cross-linker, and AIBN as an initiator. A smartphone was used as a hand-held device to take the fluorescence image of the MIPs@EDA-CDs/CF in the detection blacklight box after testing with MT in solution. Then, the ImageJ program is used to measure RGB intensity with a corresponding fluorescence image. This was the first time that cotton fabrics were used as a substrate for the MIPs@EDA-CDs to establish a simple portable sensor. In combination with smartphone-based image analysis, as a color detector, the proposed method exhibited decreasing analysis time and cost of laboratory instrumental analysis, enhancing the viability of onsite analysis. In addition, this method provided a simple preparation, quick response, selectivity, sensitivity, precision, and accuracy for MT detection in water and sediment samples.

## 2. Results and Discussion

### 2.1. Synthesis and Characterization of the EDA-CDs

#### 2.1.1. Optimum Condition for Synthesis and Characterization of the EDA-CDs

A 1000 mg/L EDA-CDs solution (300 W; 8 min) was analyzed using a fluorescence spectrophotometer to obtain the excitation spectrum and fluorescence emission spectrum, as shown in Figure 1a. From the spectrum, the maximum excitation wavelength (λ_ex_) was found at 353 nm, and the maximum fluorescence emission wavelength (λ_em_) was found at 450 nm. The exposure time for synthesizing EDA-CDs through the microwave was investigated at a fix power of 300 W. The fluorescence emission spectra of 1000 mg/L EDA-CDs solution from different exposure times (λ_ex_ = 353 nm) is shown in Figure 1b.

As shown in Figure 1b, the EDA-CDs were synthesized at the fixed power of 300 W for 8 min as the optimum condition through the highest fluorescence intensity. The EDA-CDs gave higher intensity than CDs synthesized from only CA due to using EDA as N-doping precursors. In addition, EDA also served as a surface passive agent which enhanced and stabilized the strong fluorescence emission of the EDA-CDs by trapping surface electrons for efficient radiative recombination [49]. The schematic of the reaction between CA and EDA (Figure 2) involved condensation polymerization and further carbonization [50]. As the reaction proceeded using the microwave at different exposure times, the polymer-like CDs were changed into carbogenic CDs, depending on the temperature. At a high temperature, carbogenic CDs were formed and exhibited higher fluorescence emission intensity due to combination of carbogenic core and surface electrons. The fluorescence intensity of the EDA-CDs dropped to a lower intensity, and the emission wavelength shifted to a longer wavelength when the exposure times were longer than 8 min as a consequence of being burnt off of the EDA-CDs solution as shown in Appendix A. The different fluorescence emissions from the various EDA-CDs solutions under UVA radiation are shown in Appendix A.

#### 2.1.2. Characterization of the Carbon Dots Using FTIR

As shown in Appendix A, the obtained EDA-CDs were characterized and compared with unmodified CDs using FTIR with the KBr method by scanning %Transmittance with the wavenumber ranging from 4000 to 400 cm^−1^ (32 scans and 2 resolutions). Attribution to the vibrational modes of citric acid molecules in Appendix A (blue), broad absorption features at around 3443 cm^−1^, and 1397 cm^−1^ confirmed the presence of the -OH group. In addition, the peak at 1731 cm^−1^ was observed, which was attributed to the C=O stretching of the carboxylic group of CA while C-OH stretching and CH_2_ rocking vibrations were observed at 1078 cm^−1^ and 790 cm^−1^, respectively. Compared with unmodified CDs, the peak C=O stretching at 1731 cm^−1^ disappeared from the EDA-CDs spectrum as shown in Appendix A (red). The results confirmed that CDs were modified with EDA by changing the carboxylic group to the amide group. Moreover, the strong peaks of amide C=O stretching at 1659 cm^−1^ and N-H bending vibrations at 1558 cm^−1^, in addition to C-NH-C asymmetric stretching at 1118 cm^−1^, confirmed the presence of the CONH groups on the CDs [51].

### 2.2. Preparation of the EDA-CDs Grafted Cotton Fabric (EDA-CDs/CF)

The adsorption ability of the EDA-CDs/CF was investigated by measuring the differences between the initial concentrations and remaining concentrations of EDA-CDs in a solution using a calibration method from a fluorescence spectrophotometer. To obtain the highest adsorption, the volume of EDA-CDs solutions (1000 mg/L) and sonication times were optimized using the volume from 5 to 20 mL and sonication times from 5 to 20 min. The optimum volume was chosen at 10 mL for grafting on the cotton fabric due to the highest % of adsorption, fluorescence emission, and gray intensity as shown in Appendix A. The percent of adsorption and fluorescence emission were stable at a volume greater than 10 mL because of the limitation of active sites on the cotton fabric. Likewise, the sonication time was considered from % of adsorption, fluorescence emission, and gray intensity of the EDA-CDs/CF (Appendix A). The results showed that although there were no differences in the % of adsorption, the brightest blue fluorescence emission and the highest gray intensity were found at a sonication time of 10 min. Sonication times longer than 10 min may cause carbon dots to be desorbed from the cotton fabric. 

### 2.3. Synthesis and Characterization of the MIPs@EDA-CDs/CF

#### 2.3.1. Optimum Conditions for Synthesis of the MIPs@EDA-CDs/CF

As shown in Appendix A, to obtain the highest MT adsorption as template molecules, the optimum MT concentration, MAA dosage, EGDMA dosage, AIBN dosage, and polymerization time were chosen in compliance with the highest fluorescence emission and gray intensity of the MIPs/MT@EDA-CDs/CF under UVA radiation. The optimum concentration of MT was selected at 50 mg/L and the optimum volumes of MAA, EGDMA, and AIBN were chosen at 300 μL, 700 μL, and 50 μL, respectively, with a polymerization time of 22 h.

In this work, the extraction solvent was a mixed solvent system comprising acetonitrile (ACN) and ethanol (EtOH). ACN was used as a solvent for preparation of MT solution and EtOH was used to remove unreacted reagents [52]. Therefore, the mixed solvent of EtOH to ACN ratio was optimized and chosen to extract MT from the MIPs/MT@EDA-CDs/CF, depending on the similar fluorescence emission, compared with the control NIPs@EDA-CDs/CF. As shown in Appendix A, the MIPs@EDA-CDs/CF after three time extractions with EtOH: ACN (1:4) exhibited the lowest fluorescence emission with a gray intensity of 168 ± 0.1 compared with NIPs@EDA-CDs/CF (157 ± 0.8). The remaining solution was determined using a UV–visible spectrophotometer to calculate elution efficiency. The results proved that EtOH: ACN (1:4) could be used for MT extraction with the highest elution efficiency of 98.3%. Moreover, the EDA-CDs dissolved very well in water, showing the bright blue fluorescence emission from the remaining solution under UVA radiation and the white color on the MIPs@EDA-CDs/CF under normal light as shown in Appendix A. This is the reason why water was not used for extraction.

The scheme of polymerization of the MIPs@EDA-CDs/CF by the non-covalent imprinting approach and extraction of template molecules is illustrated in Figure 3.

As illustrated in Figure 3, when the template MT was allowed to mix with monomer MAA, a non-covalent binding occurred between MT and MAA molecules. After adding cross-linker EGDMA and initiator AIBN, the polymerization started and was highly cross-linked with MT molecules attached to the functionalities of MAA [53]. Moreover, the MIPs@EDA-CDs/CF was based on the attraction between amino groups of the EDA-CDs and carboxyl groups of MAA via hydrogen bonds and van der Waals [54]. After the polymerization was finished, the MIPs/MT@EDA-CDs/CF was washed with EtOH:ACN (1:4). As a result, the specifically structured cavities of the MIPs@EDA-CDs/CF were formed from the imprint of MT molecules.

#### 2.3.2. Characterization of the MIPs@EDA-CD/CF Using SEM-EDS and FTIR

SEM analysis of raw cotton fabric revealed fat and spirally twisted ribbon-like fibers with a rough surface from impurities such as pectin and waxes, as shown in Figure 4a [55]. To prepare the cotton fabric consisting mainly of cellulose, the fabric was treated with hot NaOH to remove the impurities from the fabric, which resulted in a smoother fiber surface (Figure 4b) [56,57]. Then, the cotton fabric was soaked in H_2_O_2_ to activate the -OH group of the cellulose [54]. The SEM image of EDA-CDs/CF exhibited a slightly rough surface because the cotton fabric was grafted by the EDA-CDs, as shown in Figure 4c. In addition, the MIPs/MT@EDA-CDs/CF in the presence of MT showed a difference in morphology, as shown in Figure 4d. Fibers in the MIPs/MT@EDA-CDs/CF had a much rougher surface than the MIPs@EDA-CDs/CF (Figure 4e) and the NIPs@EDA-CDs/CF in the absence of template MT (Figure 4f). Consequently, MT molecules could be imprinted on the MIPs@EDA-CDs/CF. The success of grafting the MIPs@EDA-CDs on the cotton fabric’s surface was confirmed by SEM and EDS analysis. Over and above that, amino groups of the EDA-CDs remained on the proposed sensor even though it was washed with desorption solvent, according to Figure 4g.

The treated cotton fabric, the MIPs/MT@EDA-CDs/CF, the NIPs@EDA-CDs/CF, and the MIPs@EDA-CDs/CF were characterized using Attenuated Total Reflectance—Fourier Transform Infrared (ATR-FTIR). As shown in Appendix A, the NIPs@EDA-CDs/CF, the MIPs@EDA-CDs/CF, and the MIPs/MT@EDA-CDs/CF revealed a prominent peak at 3335 cm^−1^ due to the -NH_2_ and -OH functional groups while the treated cotton fabric showed a low intensity at 3333 cm^−1^ due to only O-H stretching. The absorption bands with a weak peak around 2900 cm^−1^ indicated the stretching of aliphatic -CH_2_- and -CH_3_ groups. Additionally, the absorption peak at 948 cm^−1^ corresponded to C-H out-of-plane bending at a double bond of MAA. The presence of two significant peaks around 1721 cm^−1^ and 1149 cm^−1^ exhibited the existence of EGDMA due to C=O stretching and C-O stretching, respectively. Moreover, the MIPs/MT@EDA-CDs/CF, the NIPs@EDA-CDs/CF, and the MIPs@EDA-CDs/CF revealed a peak at 1635–1653 cm^−1^ due to the N-H bending of the carbon dots except that the treated cotton fabric did not show any peak around these wavenumbers. These peaks confirmed that the amino group of EDA-CDs was still grafted on the cotton fabric as well as it was washed with desorption solvent. Hence, this proved that the desorption solvent did not interfere with or reduce the fluorescence on the proposed cotton sensor. In addition, the MIPs/MT@EDA-CDs/CF showed a sharp peak at 1636 cm^−1^ as α, β-unsaturated C=O stretching of the MT.

### 2.4. RGB Fluorescence Imaging of the MIPs@EDA-CDs/CF 

The treated cotton fabric, EDA-CDs/CF, MIPs/MT@EDA-CDs/CF, NIPs@EDA-CDs/CF, and MIPs@EDA-CDs/CF were taken under UV radiation and analyzed using RGB fluorescence imaging to confirm the visualized fluorescence emission on the proposed cotton sensor.

As shown in Figure 5, fluorescence images were obtained from charge-coupled device (CCD) imagers. The RGB intensity was measured from the corresponding images with fluorescence emission. Although the treated cotton fabric did not exhibit any blue fluorescence emission, RGB intensity was measured from black color and reported as shown in Table 1. The EDA-CDs/CF exhibited some fluorescence and was higher than both NIPs@EDA-CDs/CF and MIPs@EDA-CDs/CF. In addition, the MIPs/MT@EDA-CDs/CF exhibited the brightest fluorescence emission which corresponded with the highest fluorescence intensity because of templating MT. 

The enhancement fluorescence emission of the MIPs@EDA-CDs/CF after imprinting MT in MIP structure could be explained as follows. MT interacted with MAA through formation of hydrogen bonds as MT has H-bond acceptors. Moreover, MT also interacted with amino groups through hydrogen bonding or electrostatics, which can enhance the conjugation degree of amino-passivated EDA-CDs/CF. As a result, the fluorescence emission increased as shown in Figure 5c and the corresponding RGB intensity of the MIPs/MT@EDA-CDs/CF was the highest value (Table 1). After removing the MT, the fluorescence emission and corresponding intensities of the MIPs@EDA-CDs/CF and the NIPs@EDA-CDs/CF were compared and no significant differences were found. Both cotton sensors still exhibited high fluorescence emission and were weaker than that of the EDA-CDs/CF. The EDA-CDs grafted on the CF were rich in amino groups which could interact with carboxyl groups of MAA via hydrogen bonds and van der Waals. Copolymerization due to the reaction of amino groups in the EDA-CDs/CF with EGDMA might not occur during the polymerization of MAA with EGDMA. In conclusion, MIPs or NIPs were coated on the EDA-CDs/CF without chemical crosslinking.

### 2.5. Detection of MT Using the MIPs@EDA-CDs/CF with Image Processing via Smartphone and ImageJ Program

A smartphone was used to take optical images of the MIPs@EDA-CDs/CF under UVA irradiation inside the detection box at 90° for fluorometric measurement. 

These fluorescence images were then processed with the ImageJ program coupled with a personal laptop for gray, red, green, and blue measurements. The RGB values from converting the image with the corresponding gray intensity were compared with a blank to obtain ∆ gray intensity. The ∆ gray values were calculated from the blank subtraction using Equation (1). In comparison with the blank, ∆ red, ∆ green, and ∆ blue intensities with the corresponding red, green, and blue values were calculated from Equations (2)–(4).
(1)$$\Delta \;gray\;intensity\;\;=gray_{MT}-gray_{blank}$$
(2)$$\Delta \;red\;intensity=red_{MT}-red_{blank}$$
(3)$$\Delta \;green\;intensity\;\;\;=green_{MT}-green_{blank}$$
(4)$$\Delta \;blue\;intensity=blue_{MT}-blue_{blank}$$
where MT and blank were indicated as MT detection and blank detection, respectively.

The color changes on the MIPs@EDA-CDs/CF were not observed by naked-eye detection but could be observed under UVA radiation. A smartphone-based fluorescence ImageJ program was used for quantitative analysis. Under UVA radiation in a detection blacklight box, the MIPs@EDA-CDs/CF exhibited the lowest blue fluorescence emission in the absence of MT (Figure 6c), which is the corresponding fluorescence emission of the EDA-CDs solution at a wavelength of 450 nm (Figure 1). The represented optical fluorescence images of the MIPs@EDA-CDs/CF after MT detection at the concentrations of 100 μg/L, 300 μg/L, 500 μg/L, 800 μg/L, and 1000 μg/L are also shown in Figure 6c. The blue fluorescence of the MIPs@EDA-CDs/CF used as sensors for MT detection in acetonitrile solution gradually increased with the increasing MT concentration. From ImageJ analysis (Appendix A), the highest blue intensity was observed with their corresponding bright blue fluorescence emission from the modified cotton sensor. Although the highest color intensity was measured from the blue component, the relationship with MT concentration was not linear and seemed to be unchangeable. As shown in Figure 6a, the Δ gray, ∆ red, and ∆ green values gradually increased from the lower to the higher concentration of MT. However, there was no significant change when the concentration was greater than 1000 μg/L. The linearity range was observed from 100 to 1000 μg/L, as shown in Figure 6b. It was found that the linearity range of the relationship between the ∆ red intensity and MT concentration gave the highest slope and the best linearity with the correlation coefficient R^2^ = 0.999 because the highest coefficient is equal to 1. The limit of detection (LOD) and the limit of quantification (LOQ) of the ∆ red intensity were 44.4 μg/L and 134 μg/L, respectively. The LOD and LOQ were calculated based on 3.3 × S/b and 10 × S/b, respectively, where b is the slope obtained from the calibration curve, and S is the standard deviation of the response [58,59]. Consequently, the ∆ red intensity was chosen for the quantitative analysis of MT and further analysis. In this work, the proposed method was validated to demonstrate that fluorescence images taken with a smartphone and analyzed with the ImageJ program could provide trustworthy and accurate data.

### 2.6. Validation of the Proposed Method

#### 2.6.1. Selectivity of the MIPs@EDA-CDs/CF

Different ions, organic compounds, and structural analog molecules of the same concentration (500 μg/L) including Na^+^, K^+^, Ca^2+^, Mg^2+^, SO_4_^2−^, Cl^−^, phenol, nonylphenol, and testosterone were carried out to study the selectivity of the MIPs@EDA-CDs/CF towards MT detection. All interferences caused no significant differences in fluorescence emission of the MIPs@EDA-CDs/CF (0–4.9%) which was less than NIPs@EDA-CDs/CF as the control sensor (10.5–24.6%) for MT detection, as clearly shown in Figure 7a.

From Figure 7b, although both MIPs@EDA-CDs/CF and NIPs@EDA-CDs/CF responded to all interferences, they exhibited the highest ∆ red intensity for MT detection. The results proved that the MIPs@EDA-CDs/CF was highly selective towards MT compared with NIPs@EDA-CDs/CF. High ∆ red intensity for MT detection on the MIPs@EDA-CDs/CF and the NIPs@EDA-CDs/CF depended on their adsorption ability. The adsorption capacity of the MIPs@EDA-CDs/CF was about 83% higher than the NIPs@EDA-CDs/CF as a consequence of higher ∆ red intensity of about 30% for smartphone-based image detection of MT at 500 μg/L. Due to the interaction between MT and MAA in polymeric structure, MT could be adsorbed by both cotton sensors. However, the NIPs@EDA-CDs/CF could adsorb less MT than the MIPs@EDA-CDs/CF. The reason for this is that self-association of MAA occurred at the NIPs@EDA-CDs/CF as a result of a decrease in binding sites of MAA for MT adsorption. Although the extent of the high signal for detection of MT on the NIPs@EDA-CDs/CF was observed, the MIPs@EDA-CDs/CF exhibited a higher selective MT adsorption which was the main purpose of using the MIP technique.

#### 2.6.2. Reusability, Stability, and Precision of the MIPs@EDA-CDs/CF

The MIPs@EDA-CDs/CF was tested with 1000 μ/L of MT for reusability and stability. The MIPs/MT@EDA-CDs/CF was then washed with EtOH:ACN (1:4) to remove MT out and tested again with MT to investigate the ability of the cotton sensor for several cycles. 

As shown in Figure 8a, the MIPs@EDA-CDs/CF was continually tested again and again by removing MT and rebinding MT for 12 cycles. For each cycle, the upper dashed line represented MT rebinding to the MIPs@EDA-CDs/CF, corresponding with high read intensity while the lower dashed line of low red intensity exhibited an absence of MT from extraction. However, it was observed that the MIPs@EDA-CDs/CF could only be used for two cycles by considering the differences between high and low intensity levels. After two cycles, the red intensity decreased gradually with the increasing cycle usage because the cavity structures of MIPs on the proposed cotton sensor may be damaged in the washing process. The long-term stability of the MIPs@EDA-CDs/CF was investigated in a dark place at room temperature and in a refrigerator (4−8 °C) for certain periods. The reason for preserving it in a dark place was to avoid exposure to visible light, which could cause the changing optical properties of the EDA-CDs; the reason for preserving it in the refrigerator was to keep it more stable before MT detection. As shown in Figure 8c, the MIPs@EDA-CDs/CF could be kept stable for 9 days because EDA-CDs on the MIPs@EDA-CDs/CF could avoid deterioration or photochemical reaction by exposure to visible light. Whereas, as shown in Figure 8d, the MIPs@EDA-CDs/CF could be kept stable for 18 days because EDA-CDs on the MIPs@EDA-CDs/CF could avoid degradation at a proper temperature and could avoid deterioration of the optical properties of the EDA-CDs. In contrast, the MIPs@EDA-CDs/CF could be kept stable for only 4 days under normal light because EDA-CDs on the cotton sensor may decay or have a photochemical reaction by exposure to visible light, as shown in Figure 8b.

To evaluate the precision of the proposed method, three different concentrations of MT (100 μg/L, 500 μg/L, and 1000 μg/L) were analyzed on the same day (intra-day) and three following days (inter-day). The detection was found to be precise at the ppb level, with %RSD of the five replicates ranging from 1.01–5.40% on the intra-day and 1.29–5.87% on the inter-day.

### 2.7. Binding Capacity of the MIPs@EDA-CDs/CF and the NIPs@EDA-CDs/CF

The MIPs@EDA-CDs/CF and the NIPs@EDA-CDs/CF were respectively tested with 100 μg/L, 500 μg/L, and 1000 μg/L. Then, both cotton sensors were taken out and kept air-dried in a dark place. The remaining solutions were then analyzed by a UV–visible spectrophotometer.

The amounts of MT binding to the MIPs@EDA-CDs/CF and the NIPs@EDA-CDs/CF are shown in Table 2. The result showed that a large amount of specific binding sites was formed during the polymerization process, which was more activated even at high MT concentrations. As a result, the adsorption capacity and partition coefficient of the MIPs@EDA-CDs/CF increased with an increase in the initial concentration, and it was dramatically higher than the NIPs@EDA-CDs/CF. 

Furthermore, the imprinting factor measures the imprinted polymer’s strength of interaction with the template molecule. The high imprinting factor value might interact more strongly with the template molecule and have a greater binding site than non-imprinted polymers [60]. It was discovered that the interaction was positively correlated with the imprinting factor of 11.9, implying that the imprinted polymers were strongly selective to the MT.

### 2.8. Determination of MT in Real Samples

In this study, the proposed method was applied to determine MT in water and sediment samples and compared with the HPLC analysis. Spiking MT solutions at 200 μg/L, 400 μg/L, and 600 μg/L into both samples was investigated to calculate the percentage of recovery for an accuracy study.

As shown in Table 3, the percentages of recoveries were in the range of 96–102 with a %RSD ranging from 1–4 for the water samples. Additionally, the percentages of recoveries were in the range of 99–104 with %RSD ranging from 1–6 for the sediment samples. Compared with HPLC, the proposed method was a reliable technique for MT detection in both samples with a relative error of less than 5%. MT concentrations in the unspiked samples were found at 129 μg/L in the water sample and 7.30 mg/kg in the sediment sample using the proposed method. As a result, MT concentrations in the area of interest were found at a high level due to the long-term use of the pond without cleaning, demonstrating that it might be detrimental to the aquatic organisms living at the bottom of the pond, the environment, and humans.

## 3. Materials and Methodology

### 3.1. Reagents and Materials

17 α-methyltestosterone (MT, powder, >98% purity), methacrylic acid (liquid, >99% purity), ethylene glycol dimethacrylate (liquid, ˃97% purity), and ethylenediamine (liquid, 98% purity) were purchased from Tokyo Chemical Industry Co., Ltd., Japan. Azobisisobutyronitrile (12 wt% in acetone) was purchased from Sigma Aldrich, USA. Citric acid anhydrous (solid, ACS reagent, ≥99.5% purity) and ethanol (liquid, ACS grade, >99% purity) were purchased from Merck, Germany. Acetonitrile (liquid, >97% purity) was purchased from Macron and Avantor, USA. Hydrogen peroxide (30%, AR grade) was purchased from QReC New Zealand, while sodium hydroxide (pellets, AR grade, >98.6%) was purchased from Fisher, India. An SPE cartridge (C18E 500 mg/30 mL) was purchased from Welch, USA. Cotton fabric, 220 threads per inch, was supplied from a local company in Thailand. 

### 3.2. Instrumentation

The EDA-CDs were synthesized using a domestic microwave (Samsung digital microwave series MS23K3513AW; 23L, Thailand). The synthesized EDA-CDs were characterized using the F-2500 fluorescence spectrophotometer (HITACHI, Japan) with an excitation wavelength λ_ex_ of 353 nm and an emission wavelength λ_em_ of 450 nm, whereas a UV–visible spectrophotometer 1800 Double Beam (Shimazu, Japan) was used to determine the remaining MT in a solution.

Thermo Fisher Scientific Nicolet 6700, USA, provided ATR-FTIR spectra ranging from 4000–400 cm^−1^ (32 scans, 2 cm^−1^ resolution) for characterization of EDA-CDs, MIPs/MT@EDA-CDs/CF, MIPs@EDA-CDs/CF, and NIPs@EDA-CDs/CF using PIKE instruments and a single refection horizontal ATR accessory (incident angle of 45°).

The morphological analysis of the cotton fabrics was performed using a scanning electron microscope (SEM; JSM-6610 LV) obtained from JEOL, Thailand. Furthermore, an energy-dispersive X-ray spectrometer (EDS, INCA-xart, Oxford, Thailand) was used to determine the chemical composition of raw cotton fabric, treated cotton fabric, EDA-CDs/CF, MIPs/MT@EDA-CDs/CF, MIPs@EDA-CDs/CF, and NIPs@EDA-CDs/CF.

The confirmation of visualized fluorescence on the cotton sensor, treated cotton fabric, EDA-CDs/CF, MIPs/MT@EDA-CDs/CF, MIPs@EDA-CDs/CF, and NIPs@EDA-CDs/CF was taken under UV radiation using a highly sensitive and robust charge-coupled device imager (Amersham ImageQuant 800 with Epi-UV; 360 nm as a light source) for RGB fluorescence imaging. 

The comparison of the proposed method was conducted using an HPLC-UV detector (Varian Pro Star, Thai Unique Co., Ltd.) with a RP-C18 column (4.6 mm × 250 mm, particle size 5 µm). The temperature of the column was kept constant at 25 °C, and MT was detected at a wavelength of 245 nm. The mobile phase was water–acetonitrile (30:70, *v*/*v*) with a flow rate of 1.0 mL min^−1^.

### 3.3. Smartphone and Detection Box Set Up

Quantitative analysis was conducted by using the ImageJ program to measure RGB intensity. The sample was fixed in a black detection box and the smartphone (iPhone 8plus) was used to take an image of the MIPs@EDA-CDs/CF at 90 degrees under UVA radiation. The optical fluorescence images were analyzed using the ImageJ program to measure gray, red, green, and blue intensities. The detection box, smartphone and setup are shown in Figure 9 and Table 4.

### 3.4. Synthesis of Ethylenediamine-Carbon Dots (EDA-CDs)

An amount of 5 g of CA anhydrous was dissolved with 20 mL of deionized water in a 100 mL beaker, and 2.50 mL of EDA was added to the solution. After adding EDA, the reaction produced heat. Then, the mixture solution was cooled by stirring for 10 min before heating in a microwave at 300 w for 8 min (the beaker was covered by a watch glass). After that, the obtained brown EDA-CDs solution was evaporated by a rotary evaporator and completely dried in an oven at 40 °C for 6 h. Finally, the obtained brown EDA-CDs powder was collected and kept in a dark place for further use.

### 3.5. Preparation of the EDA-CDs Grafted Cotton Fabric

Cotton fabric (20 × 20 cm) was treated with 600 mL of 5% *w*/*v* NaOH for removing wax and pectin at 95 °C for 2 h. After that, the cotton fabric was washed with deionized water many times to make it neutral. The treated cotton fabric was then soaked in 250 mL of 5% H_2_O_2_ for 2 h before being rinsed with 500 mL of deionized water 3 times. Then, the treated cotton fabric was dried in the oven at 105 °C for 5 min and cut into small pieces (2 × 2 cm). After that, a piece of cotton fabric and 10 mL of 1000 mg/L EDA-CDs solution were placed into a 50 mL Erlenmeyer flask and sonicated for 10 min. Finally, the EDA-CDs grafted cotton fabric (EDA-CDs/CF) was taken out and air-dried in a dark place for further use.

### 3.6. Preparation of the MIPs@EDA-CDs/CF

An amount of 5 mL of 50 mg/L MT and 300 μL of MAA were added to the Erlenmeyer flask and shaken using a shaking incubator for 30 min. Then, 700 μL of EGDMA and 50 μL of AIBN (12% in acetone) were added to the Erlenmeyer flask. The solution was shaken and polymerized using a shaking incubator at 60 °C for 6 h. After that, a piece of EDA-CDs/CF from Section 3.5 was added to the solution system and kept for 22 h at room temperature in a dark place. After polymerization, the obtained MIPs/MT@EDA-CDs/CF was taken out, kept air-dried, and washed with 10 mL of EtOH: ACN (1:4; *v*/*v*) by a vortex mixer level 1 for 10 min. The washing procedure was repeated 3 times to completely remove the MT template molecules and unreacted reagents from the MIPs/MT@EDA-CDs/CF. To confirm that no MT was left, the remaining solutions after washing were analyzed with a UV–visible spectrophotometer and the obtained MIPs@EDA-CDs/CF was then subjected to the detection process using the smartphone-based image analysis method. After no MT was detected, the MIPs@EDA-CDs/CF was kept in a brown zip plastic bag in an airtight container before preserving it in a refrigerator for further use. The non-imprinted composite NIPs@EDA-CDs/CF was prepared as a control using an identical procedure without the addition of MT. 

### 3.7. Validation of the Proposed Method

#### 3.7.1. Selectivity of the MIPs@EDA-CDs/CF

The MIPs@EDA-CDs/CF and the NIPs@EDA-CDs/CF were tested with 5 mL of the mixture between 500 μg/L MT and interferences at the concentration of 500 μg/L (Na^+^, K^+^, Ca^2+^, Mg^2+^, SO_4_^2−^, Cl^−^, testosterone, phenol, and nonylphenol), shaken by hand for 5 min and then kept for another 10 min. After that, both MIPs@EDA-CDs/CF and NIPs@EDA-CDs/CF were taken out and kept air-dried in a dark place. The air-dried MIPs@EDA-CDs/CF and the NIPs@EDA-CDs/CF were then subjected to the detection process the using smartphone-based image analysis method.

#### 3.7.2. Reusability, Stability, and Precision of the MIPs@EDA-CDs/CF

A piece of the MIPs@EDA-CDs/CF was tested with 5 mL of 1000 μg/L MT, shaken by hand for 5 min, and kept for another 10 min. After that, the MIPs@EDA-CDs/CF was taken out and air-dried in a dark place. The air-dried of the MIPs@EDA-CDs/CF was then subjected to the detection process using the smartphone-based image analysis method. The MIPs@EDA-CDs/CF was then washed with 10 mL of EtOH:ACN (1:4) to remove the MT and then subjected to the detection and analysis processes. As mentioned above, the experiments were repeated for 12 cycles.

The stability of the MIPs@EDA-CDs/CF is also an important indicator concerning whether the material can be used in actual detection. The MIPs@EDA-CDs/CF was kept under normal light, in a dark place, and in a refrigerator (4−8 °C). Then, the materials were subjected to the detection process using the smartphone-based image analysis method every day for a period of use.

When evaluating the precision of the proposed method, it was necessary to assess the repeatability. A piece of the MIPs@EDA-CDs/CF was placed into a 30 mL PE bottle containing 5 mL of three concentrations of MT (100 μg/L, 500 μg/L, and 1000 μg/L). The PE bottle was shaken by hand for 5 min and kept still for 10 min. Then, the remaining experiments, as previously mentioned, were performed for detection and analysis. Each experiment was repeated with 5 replicates which were analyzed on the same day (intra-day) and three following days (inter-day).

### 3.8. Binding Specificity of the MIP@EDA-CDs/CF and the NIP@EDA-CDs/C

The adsorption experiment of specificity and selectivity was carried out to evaluate the binding capacity of the MIPs@EDA-CDs/CF while using the NIPs@EDA-CDs/CF as the control. The MIPs@EDA-CDs/CF or the NIPs@EDA-CDs/CF was tested with various MT concentrations of 100 μg/L, 500 μg/L, and 1000 μg/L. After performing the remaining experiments, as previously mentioned in Section 3.7, the remaining solution was measured with a UV–visible spectrophotometer. The adsorption capacity ($q_{t}$) was calculated based on Equation (5) [53,61,62]:(5)$$q_{t}=\frac{(C_{0}-C_{t})V}{m}$$
where $q_{t}$ (μg/g) is the mass of MT adsorbed per gram of the MIPs@EDS-CDs/CF, C_0_ was the initial concentration, and C_t_ is the remaining concentration after adsorption (μg/L) of MT. V (L) is the volume of the initial solution and m (g) is the weight of the MIPs@EDA-CDs/CF or the NIPs@EDA-CDs/CF.

The specificity of the MIPs@EDA-CDs/CF and the NIPs@EDA-CDs/CF was calculated using the partition coefficients of selected homologue hormones between polymers and the solution with the same volume and amount of MT as the adsorption experiment. The partition coefficient (K) was determined based on Equation (6) [63]:(6)$$K=\frac{C_{p}}{C_{t}}$$
where C_P_ (μg/L) is the concentration of MT bound by the MIPs@EDA-CDs/CF and the NIPs@EDA-CDs/CF and C_t_ (μg/L) is the concentration of MT remaining in the solution.

The imprinting factor (IF) was used to evaluate the selectivity properties of the MIPs@EDA-CDs/CF and the NIPs@EDA-CDs/CF. In batch rebinding assays, IF is affected by the analyte concentration and the solvent used. IF was carried out using 5 mL of 500 μg/L of MT in acetonitrile. IF was calculated based on Equation (7) [18,64]:(7)$$IF=\frac{K_{MIP}}{K_{NIP}}$$
where K_MIP_ and K_NIP_ were the partition coefficients of MT for the MIPs@EDA-CDs/CF and the NIPs@EDA-CDs/CF, respectively.

### 3.9. Detection of MT in Real Samples Using the MIPs@EDA-CDs/CF and Smartphone-Based Image Analysis

Water and sediment samples were collected from Samutprakan province, Thailand, and kept at 4 °C to prevent biodegradability. Therefore, the samples needed to be analyzed for no longer than 48 h, and the sediment sample needed to be dried before analysis. Solid phase extraction (SPE) was usually used to clean up interferences, impurities, and pre-concentration. In this work, SPE (C18E) was used to change the phase from water into acetonitrile. First, the SPE was pre-conditioned with 5 mL of DI water to moisturize the solid phase, and 10 mL of water sample was loaded through the SPE column. After that, 10 mL of acetonitrile was used to elute the MT at a constant flow rate of 1 mL/min. In addition, the sediment sample was dried in an oven at 60 °C for 8 h and then homogenized using a porcelain mortar and sieved using a mesh size of 200 μm. After that, 0.3 g of dried sample was extracted by 10 mL of acetonitrile using a vortex mixer level 1 for 10 min. Finally, the supernatant of the sediment sample was filtered through filter paper (Whatman No. 1). Both water and sediment samples were filtered with a 0.20 μm nylon syringe filter before analyzing with HPLC and compared with the proposed method using the MIPs@EDA-CDs/CF and smartphone-based image analysis process.

### 3.10. Fluorescence Image

The original JPG file of the two-dimensional fluorescence images from the smartphone to the personal laptop was transferred using the Telegram application. The intensity function can be used to represent an image in the RGB color model, where color is specified by expressing how much of each of the red, green, and blue components are present in the image; each component ranges from zero to 255. When all components are set to zero, the result is black; otherwise, the result is white if it all sets as 255. This means that the RGB color space may represent 16,777,216 different colors depending on the red, green, and blue values. Nevertheless, the two related realities can bring up concerns: (1) images with the same appearance can have varying pixel values and (2) images with varied appearances can still have the same pixel values [65]. Furthermore, image quality has a significant impact on image processing and analysis. Therefore, the parameters for image collection should be kept as consistent as possible such as capture device, background, light, location, and focus [66].

## 4. Conclusions

Simple techniques were used to fabricate the MIPs@EDA-CDs/CF for MT detection. The obtained MIPs@EDA-CDs/CF was examined using FTIR and SEM-EDS to confirm the success of templating MT in the MIPs@EDA-CDs on the cotton fabric’s surface and confirmed the emission of the MIPs@EDA-CDs/CF using RGB fluorescence imaging. The novel imprinting MIPs@EDA-CDs/CF had sufficient capacity and the cavity caused rapid recognition of MT molecules with better selectivity and specificity due to the adsorption capacity and imprinting factor. The fluorescence images obtained from the smartphone were simply analyzed for the gray, red, green, and blue intensity measurement using the ImageJ program for quantitative analysis of MT in water and sediment samples. Furthermore, the proposed cotton sensor exhibited better accuracy and precision and could be used twice. In addition, the ImageJ program provided trustworthy and accurate data compared with the HPLC. The proposed method proved to be convenient and effective for monitoring the MT in water and sediment samples.

## Figures and Tables

**Figure 1 molecules-27-07257-f001:**
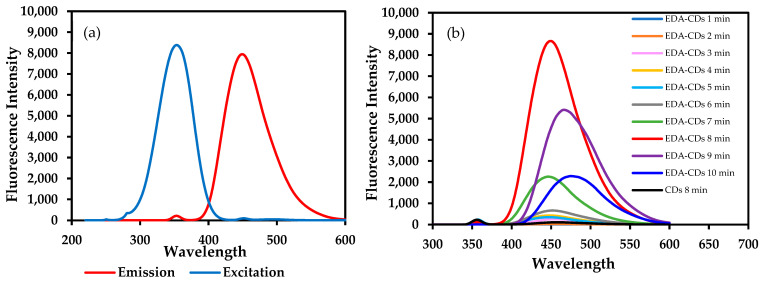
(**a**) Excitation spectrum and fluorescence emission spectrum of the EDA-CDs (300 W, 8 min) and (**b**) fluorescence emission spectra of the EDA-CDs from different exposure times, ranging from 1–10 min (300 W).

**Figure 2 molecules-27-07257-f002:**
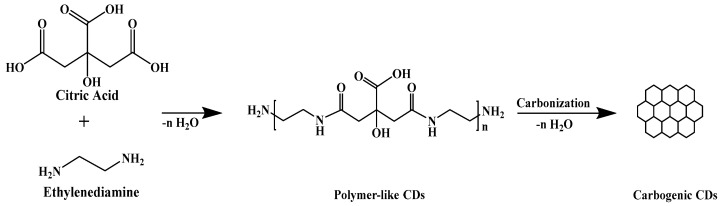
Schematic of condensation polymerization and carbonization of CA and EDA.

**Figure 3 molecules-27-07257-f003:**
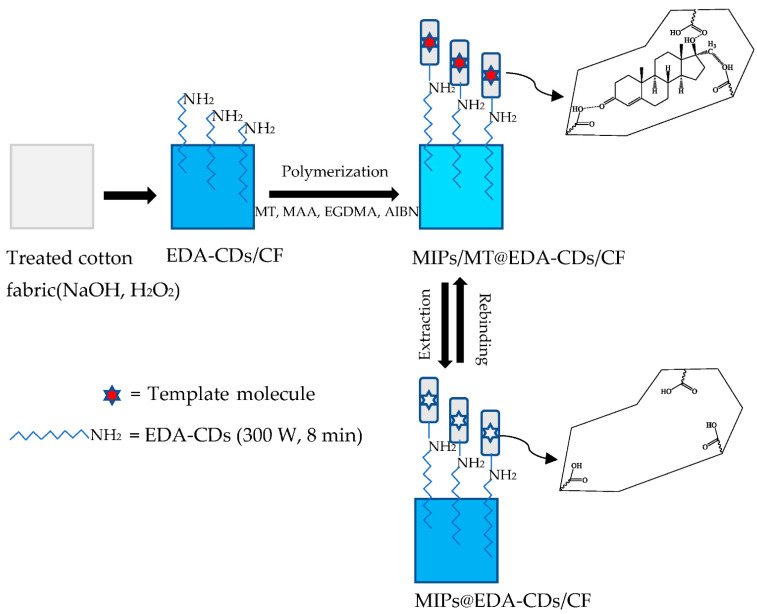
Scheme of polymerization of the MIPs@EDA-CDs/CF and extraction of the template molecule.

**Figure 4 molecules-27-07257-f004:**
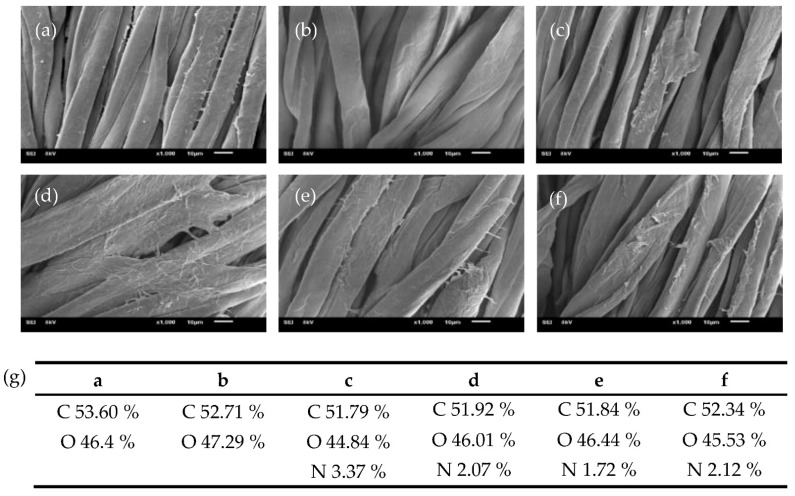
SEM images (10 μm) obtained at 8 kV acceleration voltage: (**a**) raw cotton fabric, (**b**) treated cotton fabric, (**c**) EDA-CDs/CF, (**d**) MIPs/MT@EDA-CDs/CF, (**e**) MIPs@EDA-CDs/CF, (**f**) NIPs@EDA-CDs/CF, and (**g**) EDS analysis of each cotton fabric.

**Figure 5 molecules-27-07257-f005:**
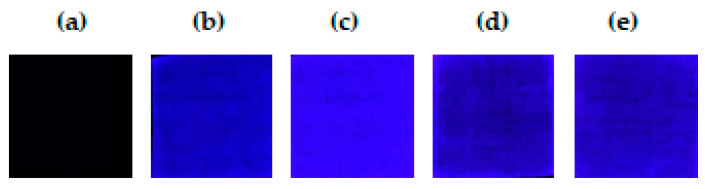
Fluorescence images of (**a**) treated cotton fabric, (**b**) EDA-CDs/CF, (**c**) MIPs/MT@EDA-CDs/CF, (**d**) MIPs@EDA-CDs/CF, and (**e**) NIPs@EDA-CDs/CF under UV radiation.

**Figure 6 molecules-27-07257-f006:**
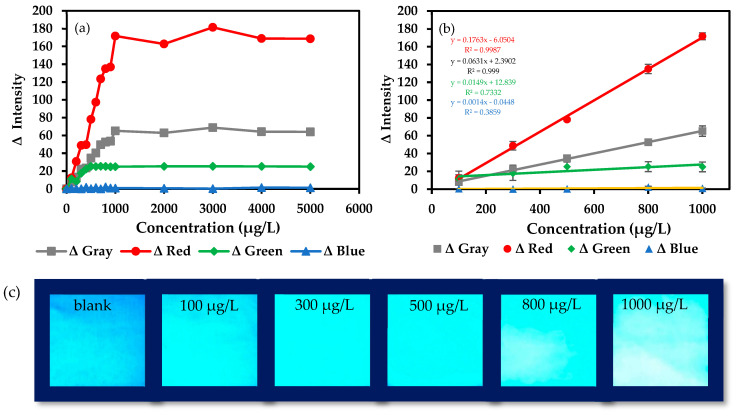
(**a**) Graph plotted between Δ Intensity and the concentrations of MT ranging from 0 to 5000 μg/L, (**b**) calibration curves showing linearity between Δ Intensity and the concentration of MT, and (**c**) the optical fluorescence images of the MIPs@EDA-CDs/CF after MT detection at different concentrations in the detection box using the proposed method.

**Figure 7 molecules-27-07257-f007:**
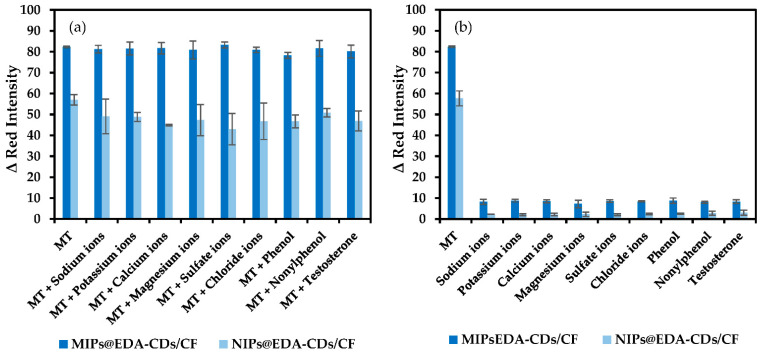
Effect of interferences on MT detection (500 μg/L) in (**a**) the binary system and (**b**) the single system using the MIPs@EDA-CDs/CF and the NIPs@EDA-CDs/CF. The concentration of all interferences was 500 μg/L.

**Figure 8 molecules-27-07257-f008:**
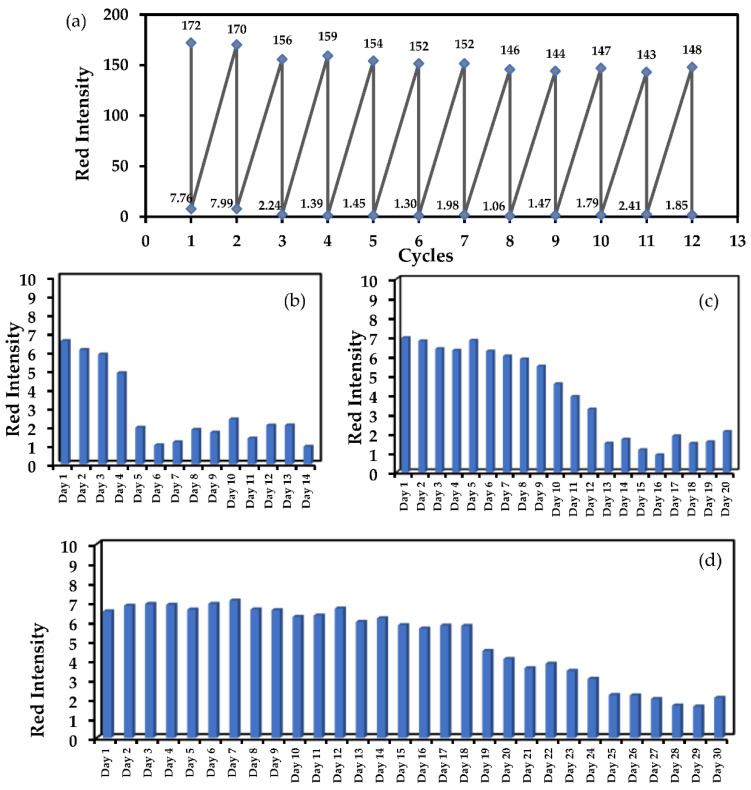
**(****a**) The cycle regeneration MIPs@EDA-CDs/CF in the absence of MT (lower dashed line) and presence (upper dashed line) of 1000 μg/L MT, (**b**) the stability of MIPs@EDA-CDs/CF under normal light, (**c**) the stability of MIPs@EDA-CDs/CF in a dark place, and (**d**) the stability of MIPs@EDA-CDs/CF in a refrigerator (4−8 °C).

**Figure 9 molecules-27-07257-f009:**
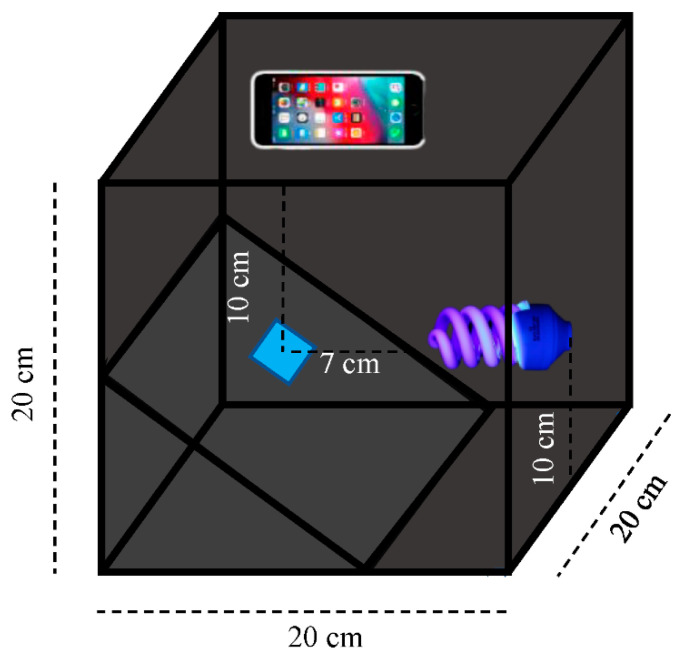
The detection blacklight box with the UVA lamp.

**Table 1 molecules-27-07257-t001:** RGB intensity of cotton fabrics: (a) treated cotton fabric, (b) EDA-CDs/CF, (c) MIPs/MT@EDA-CDs/CF, (d) MIPs@EDA-CDs/CF, and (e) NIPs@EDA-CDs/CF.

Cotton Fabrics	Area	Min Intensity	Max Intensity	Average Intensity	Std Dev
a	52440	30320	46478	38648.57	1539.39
b	52440	42342	56499	50675.78	1759.77
c	52440	46577	64139	55491.10	2208.76
d	52440	30117	48949	41239.88	1659.15
e	52440	36002	49510	42813.75	1685.95

**Table 2 molecules-27-07257-t002:** The adsorption capacity and the partition coefficient of the MIPs@EDA-CDs/CF and the NIPs@EDA-CDs/CF.

Concentration of MT (μg/L)	Sample	Adsorption Capacity (*q_t_*); (μg/g)	Partition Coefficient (K)
100	MIP 1	5.40	1.17
	NIP 1	2.99	0.40
500	MIP 2	28.1	1.19
	NIP 2	4.71	0.10
1000	MIP 3	60.7	1.24
	NIP 3	14.9	0.16

**Table 3 molecules-27-07257-t003:** A comparison between the proposed and traditional methods for MT in water and sediment samples.

Samples	Proposed Method				HPLC Found (μg/L)	%Relative Error
	Spiked (μg/L)	Found (μg/L)	%Recovery	%RSD (n = 3)		
Water	0	129 ± 4		3	134 ± 1	4
	200	320 ± 12	96	4	330 ± 2	3
	400	535 ± 6	102	1	545 ± 10	2
	600	734 ± 15	101	2	733 ± 16	0
Sediment	0	222 ± 13		6	218 ± 3	2
	200	420 ± 8	104	2	436 ± 6	4
	400	621 ± 16	100	2	611 ± 13	2
	600	814 ± 8	99	1	821 ± 16	1

n: a set of experiments using the MIPs/EDA-CDs/CF and smartphone-based image analysis.

**Table 4 molecules-27-07257-t004:** Variables of detection blacklight box and camera smartphone setup.

Detection Blacklight Box		Camera Smartphone Setup	
Variable	Value	Variable	Value
Box width	20 cm	Resolution	3024 × 4032 pixels
Box length	20 cm	Wide camera	12 MP
Box height	20 cm	Operation mode	Manual
Lamp to the cotton fabric	7 cm	Zoom	2.5
Smartphone to cotton fabric	10 cm	Flash	off
Capture angle	90 degrees	Aperture Av	f 1.8
UVA lamp	365–400 nm	Exposure Tv	1/15s
Lamp power	40 W	Quality	Raw photo
Lamp voltage	220 V		

## Data Availability

Not applicable.

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
