# Peer review of "Modification of Cotton Fabric with Molecularly Imprinted Polymer-Coated Carbon Dots as a Sensor for 17 α-methyltestosterone"

_molecules, 2022, doi:10.3390/molecules27217257_

Round 1

Reviewer 1 Report

In this work, Lim et al. proposed a smartphone-based fluorescence assay for the detection of α-methyltestosterone (MT) by grafting molecularly imprinted polymer @carbon dots onto cotton fabrics. The fluorescence images obtained from smartphone were simply analyzed for gray, red, green and blue intensity changes for the quantitative detection of MT in water and sediment samples. The proposed sensor exhibits a wide linear range and a low limit of detection, some important issues that need to be carefully addressed before publication.

(1) Why does the introduction of MT molecules into MIP@EDA-CDs/CF result in the difference of fluorescence intensity? This is an important point, and please clarify this sensing mechanism in the work.

(2) In Figure 1, what is the role of EDA in the EDA-CDs? Why is the fluorescence intensity of EDA-CDs higher than that of raw CDs? Please explain briefly the reaction between EDA and CDs.

(3) The fluorescent intensity measurement in Fig. 1(b) shows the peak is 450nm which is blue. However, the intensity change in Fig. 7 shows red channel is more obvious than blue under different MT concentrations. The authors have to explain the inconsistency.

(4) For MIP-based analysis method, the desorption process of template molecules is very important. In this work, how are the desorption solvent selected? Is the desorption solvent of ethanol and acetonitrile chosen from the reference? If so, the paper should be well cited. Moreover, what is the elution efficiency of the template molecules under optimal elution conditions?

(5) In Figure 6, there is an obvious difference of these fluorescence images among EDA-CDs/CF, MIP@EDA-CDs/CF and NIP@EDA-CDs/CF. How do the MIP and NIP affect the fluorescence intensity?

(6) In Figure 6, what is the meaning of the optical image under ultra violet? And it is not mentioned in the article.

(7) In Figure 9a, we cannot find the lower or upper dashed line mentioned in the paper. Please describe it correctly in the appropriate section. Moreover, why is the MIP@EDA-CDs/CF only used for 2 cycles? It can be observed that the red intensity of MIP@EDA-CDs/CF still remains a higher level in the presence of MT after 2 cycles.

(8) (Line 319) For the proposed sensor, the authors mentioned that the MT detection is performed in acetonitrile solution. Is the whole assay in organic solvents? If so, how is the real samples such as water or sediment samples processed before test?

Author Response

Thank you very much for all valuable comments and suggestions. Page and Line were indicated from pdf file after using Track changes (no mark up) in Word file. "Please see the attachment"

Reviewer 2 Report

The manuscript describes the development and application of a fluorescent sensor for detection of 17 α-methyltestosterone (MT).

The manuscript is well written but should be made more concise and condensed to make it more attractive for the reader. I suggest a major revision fo the paper by addressing the following comments/questions:

1. You mentioned that there are few reports on MT detection using the MIP technique (p. 3). The authors need to discuss the advantages and disadvantages of their approach, compared to already published reports based on MIP for the detection of MT.

2. At the final part of the Introduction you used two different tenses i.e.

the sentence "The MIPs were coated onto EDA-CDs to obtain MIP@EDA-CDs grafted cotton fabric (MIP@EDA-CDs/CF)." is in the past tense, while

the next sentence "The smartphone will be used as a hand-held device to take the 132 fluorescence image of MIP@EDA-CDs/CF in the detection blacklight box after testing with 133 MT in solution." is in the future tense.

Use only one tense to be concise i.e. past tense.

3. The authors need to describe the origin of signal generation. Why does the fluorescent signal increase after MT binding to the sensor? What is the expected molecular interaction of MT with CDs?

4. Move Fig. 2 and Fig. 5 into ESM.

5. The following statement is not correct: "As shown in Figure 6 and Table 1, the treated cotton fabric did not exhibit any fluorescence emission..." since treated cotton fabric exhibits an average intensity of approx 38000 units. Actually, what units did you measure fluorescence in? Can you explain the relatively high background fluorescent signal on treated cotton fabric?

6. What did you measure in Fig. 7c? provide info in the legend to this figure. 

7. In the text it is written: "The blue fluorescence of 319 MIP@EDA-CDs/CF used as sensors for MT detection in acetonitrile solution gradually 320 increased with the increasing concentration of MT.", but Fig. 7a and Fig. 7b show no or very low signal. What is then valid?

8. Discuss why you see high background signal for detection of MT on NIP. The ratio of the signal for MT observed on MIP and NIP is not particularly impressive. How can you increase such a ratio?

Author Response

(The authors gave the same response as above.)

Reviewer 3 Report

The paper describes an optic (fluorescence) cotton fabric sensors grafted with molecularly imprinted polymer (MIP) coated carbon dots for the determination of 17 α-methyltestosterone (MT). Authors describe the synthesis of the MIP, the preparation of the sensor, the characterization by different instrumental techniques such as SEM, EDS, FTIR, and the study of the sensor performance for determining the analyte by fluorescence technique using a smartphone. Finally, they apply the developed methodology to quantify MT in water and sediments samples.

The paper seems to be very interesting. However, several modifications/suggestions are recommended prior to be published in Molecules (MDPI) journal. That is why I recommend major revisions of the manuscript.

General remarks

English grammar and spelling corrections should be done throughout the whole text.

Acronyms should be revised throughout the whole manuscript, since they should be properly defined the first time they are used. For instance, in the first sentence of the Abstract, EDA is employed without being previously defined. Please, check the whole paper.

Specific remarks

Abstract (line 17): instead of ImageQuant 800 (the name of the instrument), I suggest to use the name of the technique (fluorescence). This change need to be applied to the whole text (there is even a subsection name, 2.4, with this issue). If the linear range is stated as from 0 to 1000 ppb, it sounds inappropriate and strange to mention a LOD of 41.1 ppb. I suggest to start the linear range at the lowest concentration value tested (change in the whole text too).

Introduction section: lines 64-70, the names of the instrumental techniques should not be in capitals. Screen-printed carbon electrodes are analytical tools that show very good advantages and not the disadvantages mentioned in lines 70-72, where the rest of the techniques numbered in the previous lines fit well. That is why I suggest to change the redaction in order not to include SPCE with the rest of the techniques in the list of disadvantages.

Paragraph from lines 102 to 123 should be summarized. There are many sentences stating repeated information in different parts of it. Please, rearrange it and reduced it to avoid repetitions. In the last paragraph of this section, please, enhance the novelty statement by including that the system is used for the first time, as done in the Abstract.

Results and discussion section: I would put this section after section 3. Materials and Methods (as suggested in the Guide for Authors of the journal). Hence, Results and discussion would be section 3 and Materials and methods section 2. In general, the first sentences of each subsection within Results and Discussion section could be removed, since they are repetitive: authors state there information already collected in Materials and methods section. For example, the first paragraph of section 2.1.1 contains much repeated information. Some of this information could be included in the figures caption to enrich them.

Subsection 2.1.2: FTIR peaks should be indexed in the plot. Besides, I would remove a) and b) in the caption and from the discussion in the text, since there is only one figure in the text. No need for Figure 2a and Figure 2b, only Figure 2 (red) and (blue) spectra. Modify caption as well accordingly.

Subsection 2.2: Figures S3 and S4 shows very similar values. Authors do not say anything about whether there is significant difference between the values reported. I think a significance test should be applied to these data in order to justify the optimized values.

Line 224, instead of EtOH:CAN, it should be EtOH:ACN.

Again, at the beginning of subsection 2.3.2, the first paragraph is redundant. Paragraph from line 253 to 257 could be removed from the text (already said in the Experimental part) and the information included in the caption of Figure 4.

ATR-FTIR figure: peaks could be indexed in the figure as well.

Subsection 2.4: I would change ImageQuant 800 by the name of the technique: fluorescence.

Figure 7b: fitting equations and R2 values should be included in the plot.

Paragraph at the end of page 10 and beginning of page 11 could be removed since, as in previous cases, the information reported there already appears in Experimental part. The same can be said of last paragraph of subsection 2.6 and the first one of subsection 2.7.

In Table 3, the number of measures done for each case (n value) should be clarified.

Materials and Methodology section: (the order is suggested to be exchanged with Results and Discussion section).

In general, this section can be reduced, since the same procedures are repeated many times in different subsections. I suggest to put them once and then to refer each subsection to the general procedure previously described in the corresponding subsection. For example, the measuring process is repeated several times throughout the Materials and methodology section. It should appear only once.

Subsection 3.1: the name of the reagents should not be in capitals. Subsection 3.2: information about the microwave instrument employed is not mentioned.

Subsection 3.7.1: the last sentence of the subsection could be reduced and the main information put together with the ionic interferents.

Conclusions: change ImageQuant 800 by the name of the technique; do not use the name of the apparatus.

Check all references: many of them lack of the page (number of paper) information or even the volume: 11, 13, 14, 17, 18, 32, 33, 35, 37-42, 56 and 63.

Author Response

(The authors gave the same response as above.)

Round 2

Reviewer 1 Report

This manuscript has been improved. It can be published now.

Reviewer 2 Report

The manuscript can be now accepted in the present form.

Reviewer 3 Report

After carefully revising the new manuscript, the authors have followed all the suggestions /recommendations given by the referee, hence improving the quality of their work. That is why I recommend publication of the paper in Molecules journal